# Production Review of Accelerator-Based Medical Isotopes

**DOI:** 10.3390/molecules27165294

**Published:** 2022-08-19

**Authors:** Yiwei Wang, Daiyuan Chen, Ricardo dos Santos Augusto, Jixin Liang, Zhi Qin, Juntao Liu, Zhiyi Liu

**Affiliations:** 1School of Nuclear Science and Technology, Lanzhou University, Lanzhou 730000, China; 2Brookhaven National Laboratory, United States Department of Energy Upton, New York, NY 11973-5000, USA; 3Department of Nuclear Technology and Application, China Institute of Atomic Energy, Beijing 102413, China; 4Institute of Modern Physics, Chinese Academy of Sciences, Lanzhou 730000, China; 5Frontiers Science Center for Rare Isotopes, Lanzhou University, Lanzhou 730000, China

**Keywords:** medical isotope production, accelerator, nuclear medicine, review

## Abstract

The production of reactor-based medical isotopes is fragile, which has meant supply shortages from time to time. This paper reviews alternative production methods in the form of cyclotrons, linear accelerators and neutron generators. Finally, the status of the production of medical isotopes in China is described.

## 1. Introduction

### 1.1. Definition of Medical Isotopes

Medical isotopes are radioisotopes that emit positrons or gamma rays for medical diagnosis or particulate radiation, such as alpha or beta particles for medical therapy [1].

### 1.2. Medical Use

The application process for medical isotopes is depicted in Figure 1 and can be summarized in four steps:(1a)In a reactor, irradiate a suitable target with neutrons to induce a nuclear reaction;(1b)In an accelerator, irradiate a suitable target with protons, alpha, or deuteron particles to induce a nuclear reaction;(2)Separate radioisotopes from the irradiated targets;(3)Combine the ligands with radioisotopes to prepare radiopharmaceuticals;(4)Employ the radiopharmaceuticals in nuclear medicine.

Depending on the physical characteristics of the isotopes applied, radiopharmaceuticals have different medical uses in diagnosis, therapy, or both (theranostics) [2], leading to a steady increase in the use of medical isotopes in nuclear medicine over time [3,4].

#### 1.2.1. Radiopharmaceuticals for Diagnosis

Radiopharmaceuticals are generally injected intravenously or, in some cases, taken orally [5,6]. They are transported in the blood throughout the body and, due to their high affinities with specific organs, can target different diseases, especially tumors. The γ rays emitted by radiopharmaceuticals are used for imaging. Currently, there are two main imaging applications for diagnosis in nuclear medicine: Single Photon Emission Computed Tomography (SPECT) [7,8,9] and Positron Emission Tomography (PET) [10,11,12]. The distribution of radiotracers in vivo can be detected using SPECT and PET cameras.

The main advantage of nuclear medicine diagnosis lies in its ability to find lesions earlier since diseased tissues usually first denote functional changes before later evolving into shape and structural changes [13]. Another major feature of nuclear medicine diagnosis is its ability to specifically show the locations and sizes of tumors, especially when combined with Computed Tomography (CT) or Magnetic Resonance Imaging (MRI) [14,15].

#### 1.2.2. Radiopharmaceuticals for Therapy

Therapeutic radiopharmaceuticals accumulate in diseased tissue after entering the human body. Then, their cumulative radioactive emissions can produce biological effects (e.g., killing tumor cells), which makes radiopharmaceuticals particularly suitable for cancer treatment [16]. The applications of radiopharmaceuticals for therapy include α therapy, β therapy, and Auger therapy. This review focuses on α therapy and β therapy.

#### 1.2.3. Radiopharmaceuticals for Theranostics

In theranostics, radiopharmaceuticals can be used to perform diagnostic imaging and medical treatment [17,18,19]. Imaging diagnosis is used to determine an optimal treatment modality and can help monitor and evaluate the medical treatment progress [18,20,21]. Currently, radiopharmaceuticals for theranostics use either the same radiopharmaceutical, which emits γ rays for diagnosis and α or β particles for treatment [22,23], or two different radiopharmaceuticals (one for diagnosis and the other for treatment) [24].

Radiopharmaceuticals for theranostics have developed rapidly in recent years with great progress in treating neuroendocrine tumors, thyroid cancer [20,21,25,26], prostate cancer, breast cancer [27,28], and other diseases.

### 1.3. The Status of Medical Isotope Production

Radioisotopes are divided into natural and artificial radioisotopes. Currently, there are about 200 radioisotopes in use, most of which are produced artificially [29].

With the widespread usage of radiopharmaceuticals, the stable production and supply of medical isotopes is becoming increasingly important.

Medical isotopes are generally produced via either reactors or accelerators. Typically, reactor-based medical isotopes are neutron-rich isotopes commonly characterized by a long half-life, while accelerator-based medical isotopes tend to offer a shorter half-life and usually emit positrons or γ rays [30]. Reactor irradiation is currently the most commonly used method to produce medical isotopes due to their high yield, low cost, and ease of target preparation. However, this supply is sustained by reactors that were built in the 1950–60s (Table 1). The majority of these reactors will gradually shut down before 2030.

Moreover, due to their age, and as part of the decommissioning process, reactors can be expected to have longer periods of down time due to maintenance or unplanned shutdown events for safety or technical reasons [35], increasing the risk of supply interruptions or persistent shortages. Additionally, most irradiated targets for ^99^Mo production in a reactor context use highly enriched uranium (HEU) targets that generate considerable amounts of highly radioactive waste and increase the risk of nuclear proliferation [36,37]. These factors strengthen the argument that medical isotopes produced via reactors should be replaced by accelerator-based production [38,39].

The growing interest and recent improvements in accelerator technologies have already led some medical isotopes produced via reactors to be replaced or partly replaced by accelerator-produced isotopes. There are many advantages to using medical isotopes produced by accelerators:(1)Supervision is easier, and safety is improved [40];(2)The maintenance and decommissioning costs are lower [29];(3)The amount of radioactive waste produced is less than 10% of the amount produced by a reactor, and the radiation levels are lower [41];(4)It has no risk of nuclear proliferation [42].

As shown in Figure 2, the number of cyclotrons producing radioisotopes is increasing, while the number of reactors is slowly decreasing.

## 2. Medical Isotopes

This section reviews the medical isotopes produced by cyclotrons, linear accelerators, and neutron generators and lists some of the most commonly used medical isotopes, as well as their characteristics, applications, and production methods.

### 2.1. Medical Isotopes Produced by Cyclotrons (1–5: PET Radioisotopes, 6–7: SPECT Radioisotopes, 8–10: Therapeutic Radioisotopes)

A cyclotron is a particle accelerator that accelerates charged particles and uses an electromagnetic field to get the particles to follow a spiral path to ever-increasing energies until achieving the energy necessary to produce medical isotopes via nuclear interactions [58]. Compared with linear accelerators, the beams from cyclotrons have characteristically lower beam intensity, but their energy can be higher [59]. Cyclotrons are classified according to the energy of the particles they produce. As shown in Table 2, different types of cyclotrons can produce medical isotopes for a wide range of applications.

#### 2.1.1. ^18^F

^18^F (T_1/2_ = 109.8 min) decays and emits positrons with an average energy of 0.25 MeV; hence, the distance traveled until reaching positron annihilation in tissues is short. ^18^F is the most commonly used PET radioisotope. At present, the Food and Drug Administration (FDA) has approved ^18^F radiopharmaceuticals for use in the diagnosis of a variety of diseases, such as Alzheimer’s disease, infections, and many types of cancer, as well as to evaluate treatment outcomes [61,62]. According to clinical data, [^18^F]FDG can distinguish between Parkinson’s Disease (PD), MSA with predominant Parkinsonism (MSA-P), and MSA with predominant cerebellar features (MSA-C) [63,64]. PET diagnosis is expensive and can cost over $1000, while doctors can make an early and accurate diagnosis. For that reason, the annual number of PET scans has steadily increased for many years [65]. Most ^18^F is produced via cyclotrons by exploiting two nuclear reactions:(1)^18^O (p, n) ^18^F: This reaction requires enriched (and more expensive) ^18^O target materials to produce ^18^F in a high yield [66]. Technology developments led to improvements in the target system and the production of ^18^F up to 34 GBq, as well as specific activities of 350–600 GBq/mmol 30 min after the end of bombardment [67]. Subsequently, it was found that with the irradiation of 11 MeV protons, the yield of ^18^F further increased directly with the proton current. However, the impurities also increased such that for a proton current of 20 μA, the yield of ^56^Co (4.86 MBq) and ^110m^Ag (1.51 MBq) doubled [68]. Many developing countries do not have medical isotope production facilities. If these countries desire to become self-sufficient in the production of medical isotopes, they could start by installing low-energy cyclotrons to produce ^18^F [69].(2)^20^Ne (d, α) ^18^F: This is the first production method used to produce ^18^F. This reaction is characterized by lower yields and low specific activity, so it is gradually being replaced. However, with production improvements, this method could again become an attractive alternative [70].

#### 2.1.2. ^68^Ga

^68^Ga (T_1/2_ = 68 min) is a metal PET radioisotope. Currently, there are about 100 ongoing clinical tests with ^68^Ga [61], indicating the rapid development of ^68^Ga-labelled radiotracers. Radiopharmaceuticals labeled with ^68^Ga are used for the diagnosis of neuroendocrine tumors and are highly accurate when used in patients with suspected but yet not localized neuroendocrine tumors [71]. In addition, ^68^Ga and ^177^Lu (T_1/2_ = 6.7 d) have a similar coordination chemistry, rendering them some of the most promising radiopharmaceuticals for theranostics. For neuroendocrine tumors, both [^68^Ga]Ga-DOTA-TATE and [^177^Lu]Lu-DOTA-TATE have been approved by the FDA for clinical PET diagnosis and medical treatment [72,73,74]. [^68^Ga]Ga-PSMA-11 is the first radiopharmaceutical approved by the FDA for PET imaging of PSMA-positive prostate cancer, and [^177^Lu]Lu-PSMA-617 has also been used for PSMA-targeted therapy [74,75,76,77].

^68^Ga is generally available using a ^68^Ge/^68^Ga generator and represents a relatively simple and convenient method [78] that can yield up to 1.85 GBq [79]. With the development of technology, the commercial “ionic” generators have made ^68^Ga clinically successful [80,81]. ^68^Ga obtained by generators cannot meet the growing demands, however, so the use of accelerators to obtain ^68^Ga has aroused scientific interest. Moreover, higher yields of ^68^Ga can be obtained with the ^68^Zn (p, n) ^68^Ga reaction using a small cyclotron [82,83]. The yield when using a solid target was reported as 5.032 GBq/μA·h [83]. After 6 h, impurities such as ^66^Ga and ^67^Ga only accounted for 0.51% of the total activity [84]. Compared with using a generator, this production method does not require radioactive waste treatment. Although the solid target system is complex, and the separation steps are lengthy, an automated process was developed to separate the solid target and is simpler to operate than alternative methods [85]. This nuclear reaction can also take place in a liquid target, with radiochemical and radionuclidic purities both above 99.9%. However, the yield using a liquid target was found to be significantly lower (192.5 ± 11.0) MBq/μA·h [86]. This production method using the liquid target as an alternative method still needs further optimization to improve the yield.

#### 2.1.3. ^64^Cu

Upon decay, ^64^Cu (T_1/2_ = 12.7 h) emits positrons and electrons that can be utilized for PET diagnosis and have potential applications in β therapy, thus making ^64^Cu useful as a radiopharmaceutical for theranostics. Furthermore, ^64^Cu and ^67^Cu (T_1/2_ = 61.76 h) can be radiopharmaceuticals for theranostics in order to conduct pre-targeted radioimmunotherapy [87]. Presently, the FDA has approved [^64^Cu]Cu-DOTA-TATE to localize somatostatin receptor-positive neuroendocrine tumors in adult patients. In clinical experiments, [^64^Cu]Cu-DOTA-TATE has excellent imaging quality and higher detection rates for lesions [88].

^64^Cu can be produced by small medical cyclotrons via ^64^Ni (p, n) ^64^Cu reaction with high specific activity. This production method requires an enriched ^64^Ni (at least 96%) target to obtain a high yield of 5.89 GBq/μA·h and ^64^Cu with radionuclidic purity higher than 99% [89]. The disadvantage is that the ^64^Ni target material has a low isotopic abundance (0.926%) in nature [90], meaning that the target material is expensive and must be recycled to improve its cost-effectiveness [91,92]. Alternative methods of ^64^Cu production can also be deuteron-zinc reactions such as ^nat^Zn (d, x) ^64^Cu, and ^66^Zn (d, α) ^64^Cu. Although they have lower costs, their yields are lower, and high-energy deuterons are required [93]. These factors limit actual production through such reactions.

The ^64^Ni (p, n)^64^Cu reaction is the preferred choice for ^64^Cu production in clinical applications. During the past decade, more than 20 countries, including the United States, Japan, Finland, and China, have developed ^64^Ni (p, n) ^64^Cu methods for ^64^Cu production [89,91,94], some of which are shown in Table 3.

#### 2.1.4. ^89^Zr

^89^Zr (T_1/2_ = 78.4 h) is a positron emitter and a new metal PET radioisotope ideal for immunoimaging [100]. To date, ^89^Zr-atezolizumab has been studied in renal cell carcinoma (RCC), but some obstacles were encountered, so further research is needed [101]. ^89^Zr is produced by cyclotrons involving the following nuclear reactions:(1)^89^Y (p, n) ^89^Zr: This reaction only requires low-energy protons (5-15 MeV) and targets with natural abundance ^89^Y (100%), which reduces the costs significantly. The number of interference nuclear reactions is limited; hence, one can obtain a high specific activity of ^89^Zr [102,103,104]. The yield of this (p, n) reaction can be as high as 44 MBq/μA·h under irradiation of 14 MeV protons [105]. Various methods for the isolation and purification of ^89^Zr have been proposed, including solvent extraction, anion exchange chromatography, and weak cation exchange chromatography, which can obtain ^89^Zr with high specific activity and radionuclidic purity [106]. The proton energy from small medical cyclotrons installed in hospitals can meet the requirements for bombarding the ^89^Y target, which is the main reason why many hospitals have developed ^89^Zr production processes.(2)^89^Y (d, 2n) ^89^Zr: This reaction uses low-energy deuterons (also 5–15 MeV) and has the same advantages as the aforementioned production method [102,103,104], as well as offering a higher yield of 58MBq/μA·h. However, one must still factor in the availability of the beam of particles and the costs of these two production methods [105]. Thus, more research is needed.(3)^nat^Sr (α, xn) ^89^Zr: Besides requiring α beams, if ^nat^Sr targets are used, abundant quantities of impurities such as ^88^Zr and ^86^Zr can easily be produced. For the moment, this production method is only theoretically feasible [107].

#### 2.1.5. ^124^I

^124^I (T_1/2_ = 4.176 d) is a PET nuclide that can provide a higher quality diagnostic image [108]. Currently, ^124^I is used for the clinical diagnosis of thyroid cancer [109] and neuroblastoma [110]. ^124^I and ^131^I can also be combined as radiopharmaceuticals for theranostics to treat thyroid cancer [20].

^124^I is produced via cyclotrons through two different production methods:(1)^124^Te (p, n) ^124^I: This is the main production method currently employed. Although this method offers a relatively low production rate, it can achieve high currents and use enriched targets to improve the overall yield [108]. The average yield of this reaction is 16 MBq/μA·h, and at the end of bombardment, the impurity content of ^123^I and ^125^I only reaches about 1% [111]. Dry distillation is used to extract ^124^I [112]. On the downside, the enriched ^124^Te target material costs about 10000$/g, which is relatively expensive [113].(2)^124^Te (d, 2n) ^124^I: Has a high production yield of 17.5 MBq/μA·h, however, this reaction requires a beam of deuterons, which may be difficult to obtain and can result in impurities such as ^125^I (reaching about 1.7%) [111,114].

#### 2.1.6. ^99^Mo/^99m^Tc

^99m^Tc (T_1/2_ = 6.02 h) emits single γ rays with 0.141 MeV and is mostly used in SPECT; for the diagnosis of stroke; and to examine bone, myocardium, kidneys, thyroid, salivary glands, and other organs [61,62]. The proportion of nuclear medicine diagnosis applying ^99m^Tc accounts for approximately 80% of all nuclear medicine procedures, representing around 40 million examinations worldwide every year [115]. ^99m^Tc is mainly produced using a ^99^Mo/^99m^Tc generator. Currently, ^99m^Tc can be produced by cyclotrons through the following reactions:(1)^100^Mo (p, 2n) ^99m^Tc [116,117]: This is the main production method and is optimal with a proton energy range of 19–24 MeV and a highly enriched ^100^Mo target, such that ^98^Tc, ^97^Tc, and other impurities can be reduced to a minimum. According to the experimental data, with a proton beam energy of 24 MeV, the yield of ^99m^Tc is about 592 GBq/mA·h [118]. A target irradiated with a 24 MeV proton beam at 500 μA for 12 h yielded 2.59 TBq of ^99m^Tc [119]. GE PETtrace880 machines have obtained approximately 174 GBq after 6 h [116]. To date, TRIUMF and its partners have successfully verified the feasibility of using a 24 MeV cyclotron to produce ^99m^Tc to supply the needs of all applications in Vancouver by developing a complete process based on 16, 19, and 24 MeV cyclotron production and applied the results to relevant patents [120]. Automated modules to separate ^99m^Tc from irradiated targets of ^100^Mo are under development [121]. However, the shipped distance should be considered based on the direct product and its half-life [122];(2)^96^Zr (α, n) ^99^Mo→^99m^Tc [123,124]: This production method can produce ^99m^Tc with high specific activity. However, it has a low yield, and a beam with a high current is difficult to obtain, which limits the applicability of this production method.

#### 2.1.7. ^123^I

^123^I (T_1/2_ = 13.2 h) is a γ-ray emitter that can be utilized for SPECT diagnosis. It has especially been used for the diagnosis of Parkinson’s disease, primary and metastatic pheochromocytoma, and neuroblastoma. The sensitivity and specificity of this technology are greater than 90% [125]. It also can be used for diagnosis of the thyroid, brain, and myocardium.

Presently, there are three common production routes yielding ^123^I:

(1–2) ^124^Xe (p, 2n) ^123^Cs→^123^Xe→^123^I and ^124^Xe (p, pn)^123^Xe→ ^123^I: These nuclear reactions require a medium-energy cyclotron and can obtain with a high radionuclidic purity. The yield of these reactions simulated by MCNP was 757 MBq/μA·h. Compared with the experimental data, the maximum fluctuation was about 185 MBq/μA·h [126,127]. However, due to the use of enriched ^124^Xe targets, these methods are costly [128,129].

(3) ^123^Te (p, n) ^123^I: This production method can apply a low-energy cyclotron. When enriched targets of ^123^Te (enrichment of 99.3%) were used, an ultrapure nuclide was obtained, and the yield increased from nearly 18.5 to 37GBq 30 h after EOB (end of the bombardment) [130,131,132]. This production method is also costly because of the enriched target of ^123^Te. This alternative production method was proven feasible to produce ^123^I.

#### 2.1.8. ^225^Ac

^225^Ac (T_1/2_ = 9.92 d) has a unique decay chain that can emit four α rays, causing it to be more effective in destroying tumor cells than other isotopes. Presently, the first use of [^225^Ac]Ac-PSMA-I&T in a clinical context was successful in treating advanced metastatic castration-resistant prostate cancer [133,134,135]. Additionally, the research of [^225^Ac]Ac-DOTAGA-SP for the treatment of malignant gliomas is ongoing [136].

^225^Ac can be produced with medium-energy protons via the ^226^Ra (p, 2n) ^225^Ac reaction. The yield was only about 2.4 MBq after EOB [137], moreover, its radioactive inventory is difficult to handle [137,138,139]. Production of ^225^Ac applying high-energy protons (60–140 MeV) through bombarding a ^232^Th target can produce a high yield of 96 GBq, but this yield requires high intensity and energy [140], which are not readily available. Currently, the U.S. Department of Energy Isotope Program produces ^225^Ac using a spallation-induced reaction with high-energy protons on natural thorium.

#### 2.1.9. ^211^At

^211^At (T_1/2_ = 7.2 h) emits α particles that can be utilized in α therapy [141]. Currently, ^211^At in the form of [^211^At]At-PA and [^211^At]At-ch81C6 has been studied in glioma and recurrent brain tumors [142,143]. Gothenburg (Sweden) [144] is undergoing a clinical research using [^211^At]At-MX35(Fab)_2_ to treat ovarian cancer patients, which is an alpha-emitting radionuclide with great clinical potential [145].

^211^At is commonly produced by a medium-energy cyclotron bombarding a ^209^Bi target with α particles, causing a ^209^Bi (α, 2n) ^211^At reaction to take place [146,147]. Purifying the ^211^At from the target material was either done by a wet extraction or a dry distillation. The National Institutes of Health (Bethesda, USA) produced a maximum of 1.71 GBq in one hour, while Sichuan University in China produced a maximum of 200 MBq in 2 h [148]. However, due to the product of toxic impurities such as ^210^Po, the energy of the α beam needs to be monitored [148,149].

#### 2.1.10. ^67^Cu

^67^Cu (T_1/2_ = 61.76 h) emits γ rays for SPECT diagnosis and β particles that can be used for medical treatment. Thus, ^67^Cu can be used individually or with ^64^Cu as a radiopharmaceutical for theranostics. Presently, ^67^Cu is used for the nuclear medicinal diagnosis of neuroendocrine tumors and lymphomas [150,151] and the medical treatment of lymphoma and colon cancer [152].

^67^Cu is generally produced via the ^68^Zn (p, 2p) ^67^Cu reaction. This reaction has high recovery and needs both a medium-energy cyclotron and a highly enriched target [153,154,155]. Due to the need for high-energy protons, there are only a few laboratories in the world that can produce ^67^Cu [156]. The yield of the integral physical thick target was calculated and is shown in Figure 3.

In addition to the medical isotopes mentioned above, ^11^C [158], ^13^N [159], ^15^O [160], ^86^Y [161], ^44^Sc [162,163], ^201^Tl [164], ^47^Sc [165,166], ^32^P [167], ^67^Ga [168], and other medical isotopes produced by cyclotrons have also been reported.

Cyclotrons are the main accelerator-based drivers of medical isotope production. Their output is constantly improving due to advancements in targets [169,170], research on new nuclear reactions [171,172,173], and accelerator technology developments [174,175,176], leading not only to increased yields but also to a reduction in radioactive impurities. Most medical isotopes currently produced by reactors can also alternatively be produced by cyclotrons, and the constant improvements to the medical-isotope-producing abilities of cyclotrons have contributed to the stable supply of medical isotopes.

### 2.2. Medical Isotopes Produced by Linacs

The charged particles accelerated by a linac pass through the focusing magnetic field and the linear acceleration field once without deflection [58]. Once ejected, these particles irradiate their targets to produce medical isotopes. Linac beams are characterized by high beam intensity and lower energy [59].

In terms of linacs currently used to produce medical isotopes, proton linacs can be relatively easily employed in medical isotope production. For example, proton linacs that produce PET nuclides can reduce the weight of cyclotron magnets, and some high-energy and high-fluxes proton linacs can produce therapeutic nuclides [177,178,179]. While feasibility reports on the ability of electron linacs to produce medical isotopes are common, the pulsed beams and the cross-sections of linacs can create challenges when used in practice [41,180,181,182]. There are other linacs that accelerate other charged particles; however, these linacs will not be described here.

#### 2.2.1. ^18^F

PET radioisotopes can be produced with proton linacs. The first compact proton linear accelerator in the United States for the generation of medical isotopes produces ^18^F for a local hospital [183]. Additionally, Hitachi, Ltd. and AccSys Technology, Inc. (Hitachi’s subsidiary company) also developed a proton linac to produce PET nuclides. After bombardment for one hour, 23.5 GBq ^18^F was produced, indicating that batch production of ^18^F could be achieved [177].

^18^F (T_1/2_ = 109.8 min) can also be produced by electron linacs through a photonuclear reaction ^19^F (γ, n) ^18^F, as well as other commonly used PET radioisotopes such as ^11^C (T_1/2_ = 20.38 min), ^13^N (T_1/2_ = 9.96 min), and ^15^O (T_1/2_ = 122 s). When using a photonuclear reaction to produce these PET radioisotopes, the yields are generally lower since the cross-section is 1–2 orders of magnitude lower than that under a proton reaction. However, photonuclear reactions can use a natural target of ^19^F, thus providing lower costs compared to proton reactions [177]. Many feasibility reports on producing PET nuclides via photonuclear reactions have been published, but actual production still needs further study.

#### 2.2.2. ^99^Mo

^99^Mo (T_1/2_ = 66 h) decays into ^99m^Tc (T_1/2_ = 6.02 h). An electron linac can be utilized to produce ^99^Mo via the photonuclear reaction ^100^Mo (γ, n) ^99^Mo [184,185,186]. It was reported that the yield of ^99^Mo obtained after 6.5 days of continuous bombardment of a 6 g high-purity ^100^Mo target with 36 MeV electrons was 458.8 GBq (average beam power of ~8 kW) [187]. The cost of this production method can be reduced by using a natural target and, although this method will produce the isotopes of Mo, isotopes of Tc will not be produced, making it easy to separate ^99^Mo via chemical difference or evaporation temperature difference [188]. NorthStar and its partners have studied this production method and listed it as the main ^99^Mo supply option in their long-term plans [187]. Canadian Light Source (CLS) and TRIUMF also conducted feasibility research on this production method and plan to put it into production [189,190].

In addition to the medical isotopes mentioned above, the production of ^67^Cu [191,192,193], ^64^Cu [194], ^225^Ac [195,196], ^68^Ga [197], ^111^In [181], ^177^Lu [198], ^47^Sc [199], and other medical isotopes through linacs have been reported.

Overall, linacs have some disadvantages in terms of their design and yields [41,182,200,201]. As a backup method for the production of medical isotopes, linacs still require further research.

### 2.3. Medical Isotopes Produced by Neutron Generators

A neutron generator is an accelerator-based neutron source device that is capable of delivering neutrons through nuclear fusion reactions. These neutrons will, in turn, irradiate the target to produce medical isotopes. The nuclear fusion reactions commonly used to produce neutrons are shown in Table 4.

#### 2.3.1. ^99^Mo/^99m^Tc

^99m^Tc (T_1/2_ = 6.02h) can be produced by neutron generators [207,208]. After neutron moderation, neutrons with a specific energy can be obtained and then used to produce ^99m^Tc via the nuclear reaction of ^235^U (n, f) ^99^Mo→^99m^Tc. The advantages of this production method include both ease of supervision and overall safety, but the yield will be 1–2 orders of magnitude lower than that produced by a reactor [209]. SHINE and Phoenix Laboratory used a DT neutron generator to bombard UO_2_SO_4_ to produce ^99^Mo. After irradiation of a 5 L UO_2_SO_4_ solution for about 20 h, the yield of ^99^Mo was 51.8 GBq [210]. The disadvantage of this production method is that a long-term, stable, and high-intensity beam is difficult to achieve [211].

In addition, ^99^Mo can be produced via the nuclear reactions of ^98^Mo (n, γ) ^99^Mo and ^100^Mo (n, 2n) ^99^Mo, both of which use Mo targets instead of U targets. Additionally, sufficient activity of ^99^Mo can be produced in principle [207,208,212]. The yields of these two nuclear reactions can be increased by improving the fluxes of neutrons and the irradiation time and/or using highly enriched targets, in addition to other methods [213]. However, ^99^Mo from an irradiated ^98^Mo/^100^Mo target is a carrier-added product with a low specific activity. The biggest challenge for this method is how to develop a new type of ^99^Mo/^99m^Tc generator that meets medical requirements.

#### 2.3.2. ^67^Cu

^67^Cu (T_1/2_ = 61.76 h) is generally produced by cyclotrons. Kin proposed using neutrons to produce ^67^Cu [212]. Presently, using neutron generators via the D-T reaction in the form of ^67^Zn (n, p) ^67^Cu can produce ^67^Cu. Due to the developments of neutron generators, ^67^Cu can be produced in the hospital without the need to transport the isotope over long distances. This production method does not produce a large number of impurities [156,214], and the activity can reach hundreds to thousands of MBq [212]. However, when dealing with radioactive isotopes with GBq, the radiation facility will result in higher costs [212].

In addition to the medical isotopes mentioned above, ^89^Sr [215,216,217], ^64^Cu [218], ^47^Sc [219], ^132^Xe [220], ^225^Ac [212], and other medical isotopes produced by neutron generators have also been reported.

As a neutron source, a neutron generator is essential to produce neutron-rich medical isotopes. Although such generators have the advantages of low cost and target reusability [212,221], providing continuously high fluxes of neutrons and engaging in separation-extraction of the medical isotopes remain challenging topics [221]. Despite these challenges, generators are presently regarded as a viable alternative to the reactor-production method.

## 3. The Status of Medical Isotope Production via Accelerators in China

### 3.1. Available Accelerators for Medical Isotope Production in China

Currently, there are about 160 PET small medical cyclotrons for the routine production of ^11^C, ^18^F, and other medical isotopes to meet clinical demands in China [222]. Additionally, there are several medium- and high-energy accelerators used for medical isotope production in China.

The Chinese Institute of Atomic Energy (CIAE) and Shanghai Ansheng Kexing Company each have a C-30 cyclotron with adjustable proton energy of 15.5–30 MeV and beam currents up to 350 µA. These can be used to produce medical isotopes such as ^11^C, ^18^F, ^64^Cu, ^68^Ge, ^89^Zr, ^123^I, ^124^I, and ^201^Tl. CIAE has a 100 MeV proton cyclotron (C-100) with a beam current up to 200 µA capable of producing ^67^Cu, ^225^Ac, and other medical isotopes of interest.

The Sichuan University owns a cyclotron capable of delivering beams of protons, as well as alpha and deuteron particles (p–26 MeV and α–30 MeV).

The Chinese Academy of Sciences Institute of Modern Physics built a 25 MeV superconducting proton linear accelerator with an intensity in the order of milliamps. At present, the linac can accelerate various beams such as proton beams, ^3^He^2+^ beams, and ^4^He^2+^ beams. The energy of ^3^He^2+^ beams can reach 36 MeV at an intensity of 200 µA, while the energy of ^4^He^2+^ beams can reach 32 MeV with a current of 100 µA. The accelerator can meet the needs of medical isotope production and produce various radioisotopes such as ^99^Mo/^99m^Tc, ^117m^Sn, ^211^At, ^55^Fe, ^73^As, ^225^Ac, ^109^Cd, ^88^Y, and ^75^Se.

Lanzhou University has been instrumental in the development of advanced ion source selection, ion beam extraction, and acceleration system design, as well as target system design. Additionally, the university independently built a series of neutron generators based on D-D and D-T reactions [223].

### 3.2. The Status of Medical Isotope Production via Accelerators

There is a solid research foundation for accelerator-based medical isotope production in China. In the 1980s, Sichuan University and others successfully developed production technology for medical isotopes such as ^211^At, ^123^I, ^111^In, and ^201^Tl by relying on domestic cyclotrons and a CS-30 cyclotron [224]. Since the 1990s, CIAE has produced medical isotopes such as ^18^F, ^111^In, and ^201^Tl using a C-30 cyclotron.

In the last two decades, with the popularization and rapid development of domestic nuclear medicine, the amount of PET equipment increased to 427 by 2019. Today, 117 hospitals equipped with small medical cyclotrons routinely produce ^18^F to meet clinical needs, with an annual consumption of more than 1850 TBq. Additionally, some emerging isotopes such as ^64^Cu, ^89^Zr, and ^123/124^I have been rapidly developed for medical applications. In 2007, CIAE cooperated with Atom Hitech to carry out research on ^123^I production using enriched ^124^Xe gas at 111 GBq for each batch with a C-30 cyclotron. In 2012, Atom Hitech produced carrier-free ^64^Cu with enriched ^64^Ni at 37–74 GBq for each batch based on a C-30 cyclotron. In 2016, Sichuan University bombarded an ^89^Y target with 13 MeV protons and obtained ^89^Zr with a radionuclidic purity of more than 99% [16]. However, due to the limited availability of high-energy particle accelerators for the production of therapeutic nuclides such as ^67^Cu, ^225^Ac, and ^223^Ra, China is significantly lagging behind the advanced international levels of development. In 2021, for the first time, CIAE obtained around 22.2 MBq of ^225^Ac with radionuclidic purity greater than 99% using a C-100 cyclotron.

## 4. Summary

Presently, cyclotrons remain the primary facilities for accelerator-based medical isotope production, although linacs and neutron generators are rapidly becoming a viable alternative.

Cyclotrons with adjustable energy ranges or medium energy can produce various kinds of medical isotopes and can cover most radiopharmaceutical production needs in a region [59]. Yield and purity improvements in medical isotopes and the overall cost of cyclotron production have led researchers to explore further possibilities, including proton linacs, which have significant advantages in providing proton beams in the order of tens to hundreds of MeV [179]. These linacs can be developed in research institutes or laboratories conducting scientific experiments and physical research at the same time. For electron linacs, the cross-section of photonuclear interactions is relatively low, which restricts their practical applications. Other factors, such as impurity products and economic costs, also play major roles when evaluating production techniques and methodologies. Attempts to produce medical isotopes through neutron generators are promising and could theoretically yield the medical isotopes that are currently produced by reactors. However, improving the neutron flux rate remains a major consideration.

As medical isotopes produced by reactors often face supply shortages, interest in the use of accelerator-based techniques to produce medical isotopes will increase. We hope to develop an accelerator with the right energy, right beam types, right location, and good shielding facilities, which will play an important role in the supply of medical isotopes.

## Figures and Tables

**Figure 1 molecules-27-05294-f001:**
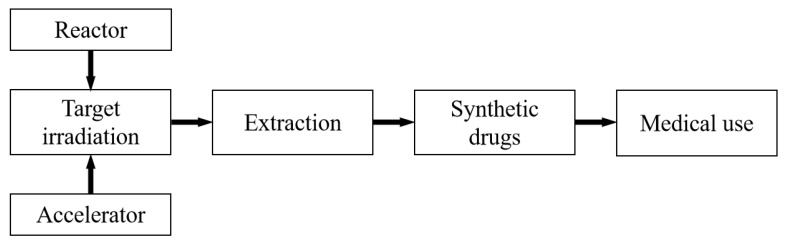
Process for the application of medical isotopes.

**Figure 2 molecules-27-05294-f002:**
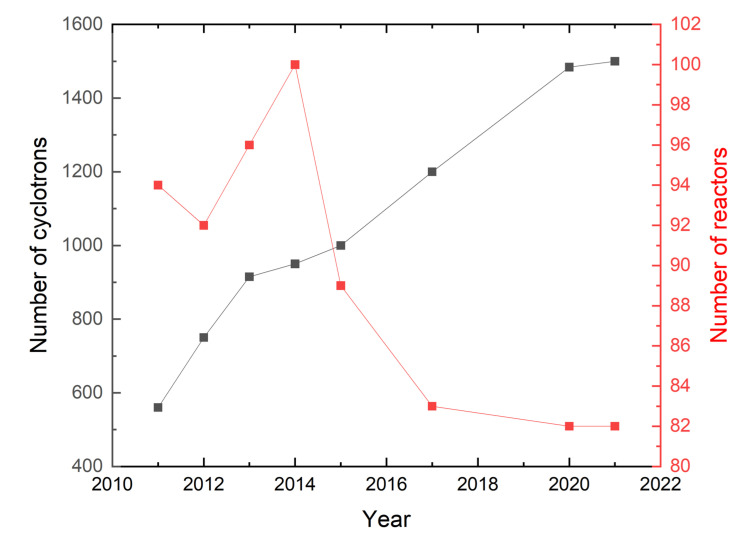
A comparison of the number of cyclotrons in the world and the number of reactors reported by the IAEA [43,44,45,46,47,48,49,50,51,52,53,54,55,56,57].

**Figure 3 molecules-27-05294-f003:**
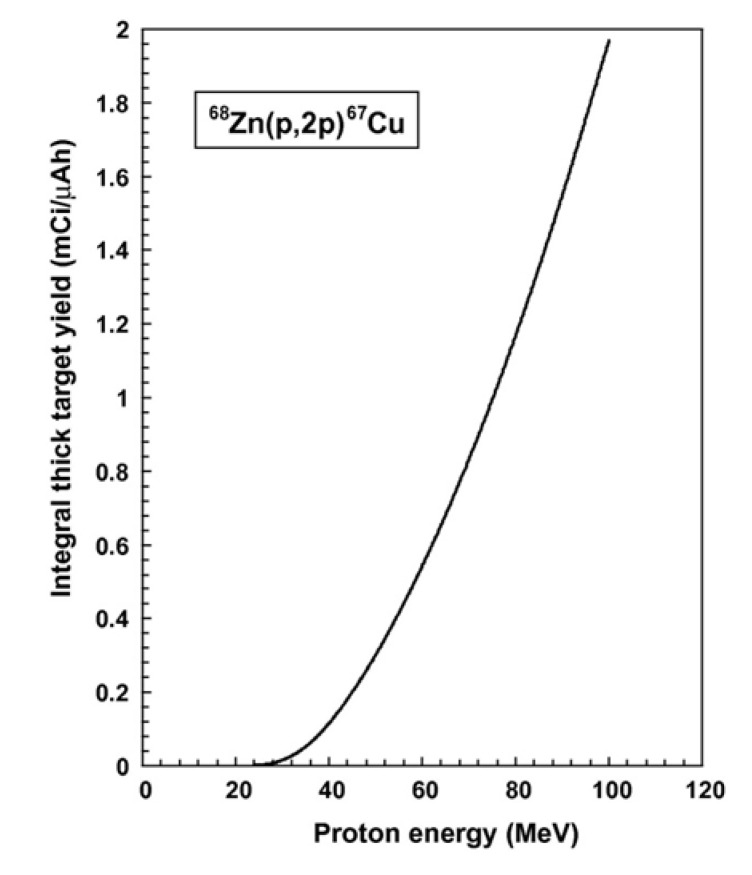
Integral physical thick target yields for the ^68^Zn (p, 2p) ^67^Cu reaction [157].

**Table 1 molecules-27-05294-t001:** Information on the world’s major reactors producing medical isotopes [31,32,33,34].

Country	Reactor	Power [MW]	Year of First Criticality	Estimated Retirement Time
Belgium	BR-2	100	1961	2026
Netherlands	HFR	45	1961	2024
Czech Republic	LVR-15	10	1957	2028
Poland	MARIA	20	1974	2030
South Africa	SAFARI-1	20	1965	2030
Russia	WWR-TS	15	1964	2025
United States	HFIR	100	1965	2035
Australia	OPAL	20	2006	2057
Germany	FRM-II	20	2004	2054

**Table 2 molecules-27-05294-t002:** Classification of medical cyclotrons [60].

Type	The Energy of Particles [MeV]	Application
Small medical cyclotron	<20	Short-lived radioisotopes for PET
Medium-energy cyclotron	20–35	Production of SPECT and some PET radioisotopes
High-energy cyclotron	>35	Production of radioisotopes for therapy

**Table 3 molecules-27-05294-t003:** Facilities that have reported the production of ^64^Cu [91,94,95,96,97,98,99].

Facility/Location	Nuclear Reaction	Irradiation Parameters	Yield
Fukui Medical University	^64^Ni(p, n)^64^Cu	12 MeV, (50 ± 3) μA	2-24 GBq in 2 h
The University of Sherbrooke PET Imaging Centre	^64^Ni(p, n)^64^Cu	15 MeV, 18 μA	3.9 GBq in 4 h
IBA	^64^Ni(p, n)^64^Cu	10 MeV, 12 μA	5123 MBq in 3 h
Paul Scherrer Institute	^64^Ni(p, n)^64^Cu	11 MeV, 40–50 μA	Max 8.2 GBq in 4–5 h
Turku PET Centre	^64^Ni(p, n)^64^Cu	15.7 MeV, < 100 μA	Max 9.4GBq after purification
Sumitomo HM-20 cyclotron	^64^Ni(p, n)^64^Cu	12.5 MeV, 20 μA	7.4 GBq in 5–7 h
NIRS AVF-930 cyclotron	^64^Ni(p, n)^64^Cu	24 MeV HH^+^, 10 eμA	5.2-13GBq in 1–3 h

**Table 4 molecules-27-05294-t004:** Fusion reactions that produce neutrons [202,203,204,205,206].

Reaction	Energy [MeV]	The Suitable Reaction of Isotope Production
D-D reaction	2–3	(n, γ)
D-T reaction	14–15	(n, 2n) (n, p)
D-^7^Li reaction	10&13	(n, 2n) (n, p)

## Data Availability

No new data were created or analyzed in this study. Data sharing is not applicable to this article.

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
