# Peer review of "Production Review of Accelerator-Based Medical Isotopes"

_molecules, 2022, doi:10.3390/molecules27165294_

Round 1
Reviewer 1 Report
The review includes various ways of production of medical radionuclides that gives general information about this area. They are briefly described and it is quite possible that this is enough for reader. However, for me this version of review lacks discussion about different reactions and ways for obtaining of definite radionuclides. Pros ans cons from various points would make it more useful, not only nude facts but also some trends and some evaluation/ gradation from authors' opinion. For example, in the subsection about 64Cu known facts are decribed without present time situation including clinical, economical features. Lack of comparison between novel or alternative methods/ reactions and already applied for clinical/ preclinical experiments also canches the eye.
Additionally, I would recommend authors to try to select basic requirements on their opinion for accelerators to be successfully used for production of one or another medical radionuclide.
Some of these comments are already introduced inside an attached file of manuscript.

Author Response
Dear Reviewer 1,
We appreciate you so much for your constructive comments on our paper. We carefully went through all of the comments and made point-to-point revision. Please be kindly noted to find the attached edition. Please let us know if any further concerns or comments.
Cheers,
Zhiyi Liu.

Reviewer 2 Report
Below is not an exhaustive list of corrections/suggestions to improve the English. I suggest using a qualified proof reader to improve the written language of the manuscript.
Line 19 - Poor english (mixing past and perfect tenses)
Line 22 - Poor english (plurals not used)
Line 26 - Expound on what radiation is suitable for use within nuclear medicine
Line 32 - Targets are not irradiated with a reactor or accelerator but in a reactor with neutrons or protons.
Line 39 - "Radiopharmaceuticals combined with different radioisotopes" - I think "Pharmaceuticals combined with different radioisotopes to produce radiopharmaceuticals" is more correct since it is not a radiopharmaceutical until it is labeled.
Line 42 - Injected intravenously?
Line 43 - Which part of the body?
line 44 - What is considered abnormal and why do they absorb radiopharmaceuticals? How do they absorb them? Why more so than healthy tissue?
Line 78 - a shorter half-life
Table 1 - Year of first criticality
Poor English used in Lines: 90, 92, 93, 94, 116, 118 - 121, 151, 163, 211, 239, 240, 248, 272-273, 303, 368, 389, 390, 392, 393,
Line 131 - O(p, n) - space missing
Half-life of Ac-225 is not 10 d. Check latest literature for 9.92 d
Lines 253 - 254, 296, 331, 347-348, 350, 359-360, 369-370 - Punctuation errors
In addition to the written corrections I would like to see more information about the amounts of each isotope being produced by each route (GBq or TBq amounts) and a short comment on how each isotope is purified from the target. Even though this is a review on the current status within China I would like to see this compared to the global production or of that within the EU or USA. The text states where the accelerators are within China and what they are capable of producing but isn't clear as to what exactly they do produce and how much. This would be of interest to readers from outside of China.
Author Response
Dear Reviewer 2,
We appreciate you so much for your constructive comments on our paper. We carefully went through all of the comments and made point-to-point revisions. Please check the attachment. Please let us know if any further concerns or comments.
Cheers,
Zhiyi Liu.

Round 2
Reviewer 2 Report
Revision is of much better quality than initial submission. Well done. Nice additions to the text. Minor comments/corrections.
Line 43: intravenously means in the vein - remove in the vein.
Line 63: An imaging diagnosis...
Table 1: HFIR at ORNL is missing as a major reactor used in the production of medical isotopes (Ac-227, W-188, Sr-89, ...)
Line 122: may induce a nuclear reaction ...
Lines 135, 187, 189, 285: 18F-FDG-PET, 64Cu-DOTA TATE and 225Ac-PSMA-I&T are the incorrect nomenclature according to https://doi.org/10.1016/j.nucmedbio.2017.09.004
Line 161: have a similar coordination chemistry which
Line 252: with a proton beam energy of 24 MeV, the yield
Line 292: Currently the US Department of Energy Isotope Program produces Ac-225 using a spallation induced reactions with high-energy protons on natural thorium
Line 368: and plan to put it into production
Line 403: yttria or yttrium oxide - not sure if all readers will know yttria
Line 467: radionuclidic purity
Author Response
Dear Reviewer 2,
We appreciate you so much for your comments and corrections on our paper. We carefully went through all of them and revised them through the language editing service. Please see the attachment and the new version.
cheers,
Liu Zhiyi.
